# Do Not Forget about Hormonal Therapy for Recurrent Endometrial Cancer: A Review of Options, Updates, and New Combinations

**DOI:** 10.3390/cancers15061799

**Published:** 2023-03-16

**Authors:** Vincent M. Wagner, Floor J. Backes

**Affiliations:** Division of Gynecologic Oncology, Department of Obstetrics and Gynecology, The Ohio State University Comprehensive Cancer Center, James Cancer Hospital, Columbus, OH 43210, USA

**Keywords:** recurrent endometrial cancer, hormonal therapy, progesterone, SERM, aromatase inhibitors, targeted therapy, mTOR inhibitor, CDK4/6 inhibitor

## Abstract

**Simple Summary:**

Endometrial cancer is the most common gynecologic cancer in the developed world and in the recurrent setting has a dismal prognosis. As the endometrium is a hormonally active tissue, hormonal therapy has long been an integral treatment modality. There are many options for hormonal therapy including progesterones, agents that target the estrogen pathway, and combinations with targeted therapies, which are described in detail in this review. High disease control rate, long duration of response, and easy tolerability make hormonal therapy an excellent option for patients with low-grade hormone receptor-positive disease who would like to avoid or cannot tolerate cytotoxic chemotherapy or other targeted therapies.

**Abstract:**

Hormonal therapy has long been a treatment modality for recurrent endometrial cancer. It is appealing for patients with low-grade, slow-growing tumors or in patients for which other treatment types may be too toxic. Hormonal therapy is well tolerated and has response rates ranging from 9 to 33%. Hormonal treatment options take advantage of the estrogen-dependent molecular pathways in endometrial cancers. Current options for hormonal therapies include progesterone therapy (medroxyprogesterone acetate and megestrol acetate) as a single agent or in combination and agents that target the estrogen pathway. Aromatase inhibitors have had modest single-agent activity, but synergistic effects have been found when used in combination with targeted therapy including mTOR inhibitors and cyclin-dependent kinase 4/6 (CDK4/6) inhibitors. Molecular profiling of endometrial cancers has begun to help individualize treatments. This review will report on existing data and ongoing trials investigating novel hormonal therapy agents.

## 1. Introduction

Endometrial cancer is the most common gynecological cancer in the developed world, and it is one of the few cancer types that is not only increasing in incidence, but also mortality [1]. Recurrent endometrial cancer carries a dismal prognosis and represents an ongoing area of unmet need for therapeutic options. Hormonal therapy has long been a treatment modality in the management of endometrial cancer. Given the recent advancement of immunotherapy for recurrent endometrial cancer [2,3], hormonal options are now most commonly used in later lines of therapy for patients with low-grade, indolent tumors or in patients for which other treatment types may be too toxic [4].

The endometrium is a hormonally active tissue. Unopposed exposure to estrogen leads to increased mitotic activity in endometrial cells, resulting in more frequent replication errors and subsequent accumulation of somatic mutations [5,6]. These mutations manifest clinically as endometrial hyperplasia and cancer [7]. This pathophysiologic relationship is the backbone of hormonal treatment for recurrent endometrial cancer. Although this relationship is strongest and most straightforward in type I tumors (low-grade endometrioid), hormones still play a role in higher risk, type II, endometrial cancers [8,9]. Type II tumors include high-grade endometrioid tumors, serous tumors, clear cell tumors, carcinosarcomas, and tumors with mixed histology. A significant portion of all endometrial carcinomas express both estrogen receptors (ER) and progesterone receptors (PR) [10].

Molecular analysis of endometrial cancer has led to further classification of the disease beyond just type I and type II. The Cancer Genome Atlas (TCGA) project identified four prognostically distinct molecular subtypes of endometrial cancer which include: ultramutated, microsatellite instable, copy-number high, and copy-number low [11]. In order to help the adaption and implementation of molecular classification into clinical practice, a pragmatic approach to classify endometrial cancer has been developed [12,13] and is now recommended by the World Health Organization [14]. This pragmatic method uses a combination of next-generation sequencing for the detection of POLE mutations and immunohistochemistry (IHC) to evaluate mismatch repair proteins and p53 status to classify tumors into the TGCA-analogous subgroups POLEmut (POLE mutations), MMRd (mismatch repair deficient), p53abn (p53 abnormal), and NSMP (no specific molecular profile). Tumors from each of the molecular subgroups have been found to express relatively high levels of ER; the highest expression is found in the NSMP subgroup and the lowest in the p53mut (which is predominantly type II or “serous-like” tumors) subgroup [12]. The expression of these hormonal receptors has been associated with tumor characteristics such as grade and stage [15]. Furthermore, there is some evidence that hormonal receptor expression is associated with better outcomes [10,16,17], however, this mechanistic relationship has not been fully elucidated [18]. Recent data suggest that within the NSMP subset of endometrial cancer, patients with ER+ disease have a very low risk of cancer recurrence [17]. Given what we know regarding the hormonal association with both the histologic and molecular subtypes of endometrial cancer, we cannot forget about the potential for hormonal therapies in the treatment of recurrent endometrial tumors.

Progestational agents have long been the prototypical hormonal therapy [19]. However, the number of hormonally based therapeutic options for patients with recurrent endometrial cancer continues to expand [20]. Some of the most novel combination options take advantage of relevant downstream driver mechanisms of ER-positive endometrial cancers. The most well-studied downstream pathway in endometrial cancer is the phosphatidylinositol-3-kinase (PI3K)/Akt and mammalian target of rapamycin (mTOR) [21,22]. This pathway has been found to be frequently upregulated in endometrial cancer by two main mechanisms: ER activation of PI3K [15,23] and direct mutation of the tumor suppressor phosphatase with tensin homology (PTEN), which is found frequently in endometrial tumors [22]. This activation of mTOR results in increased growth and proliferation; therefore, the inhibition of mTOR has gained popularity and regulatory approval in the treatment of endometrial cancer.

A second, further downstream target that has been studied extensively in breast cancer and is now being evaluated in endometrial cancer are the cyclin-dependent kinases 4 and 6 (CDK4/6). In addition to direct activation by ER, both the PI3K/Akt/mTOR pathway and a second pathway that is commonly altered in endometrial cancer [11], the receptor tyrosine kinase (RTK)/RAS/β-catenin (CTNNB1) pathway, can activate CDK4/6, leading to cell cycle progression via retinoblastoma (RB1), therefore causing cell proliferation and growth [24,25]. The inhibition of CDK4/6 has gained attention recently with the published results of two clinical trials in the treatment of endometrial cancer.

Given the rapidly changing landscape of the classification and management of endometrial cancers, it is important to review the current and future literature on hormonally relevant biomarkers and treatment regimens for advanced or recurrent disease. In preparing for this review, PubMed was searched using the following terms: endometrial cancer, hormonal therapy, endocrine therapy, progesterone, estrogen receptor modulator, and aromatase inhibitor. Results were then filtered to include the last 30 years (1993–2023). Phase II and III clinical trials in patients with advanced or recurrent endometrial cancer were included for review. Clinical trials were supplemented with the relevant literature on pathophysiology. ClinicalTrials.gov was then searched for the same terms to find ongoing or future studies. In this review, we aim to summarize the most relevant literature for hormonal therapy in endometrial cancer, including single-agent and combinations of progesterones, selective estrogen receptor modulators and degraders, aromatase inhibitors, as well as hormonal combinations with other targeted therapies. Additionally, we will discuss the current literature regarding hormonal receptor status and molecular testing as it relates to hormonal therapy. Lastly, we aim to preview the ongoing hormonally based clinical trials to give insight into some of the promising therapies of the future.

## 2. Progestational Agents Alone

When used alone, progestational agents such as medroxyprogesterone acetate (MPA) and megestrol acetate (MA) have shown modest activity in advanced or recurrent endometrial cancer. The most contemporary studies assessing single-agent progesterone include two studies by the Gynecologic Oncology Group (GOG). The first evaluated high-dose megestrol acetate (800 mg/d) in 63 patients with advanced or recurrent endometrial cancer that had either failed to respond or was considered incurable with local therapy and had not received prior cytotoxic or hormonal therapy; MA was found to have an overall response rate (ORR) of 24% with a clinical benefit rate (CBR) of 46%. The median progression-free survival (PFS) and overall survival (OS) were 2.5 and 7.6 months, respectively. Although the response rate of patients with low-grade tumors (11 of 30, 37%) was significantly higher than that of patients with high-grade tumors (2 of 24, 8%), there was no difference in response rates between cell types, including serous tumors. Grade 3 and 4 toxicities were rare (weight gain and hyperglycemia). Three deaths secondary to cardiovascular events were possibly related to therapy; diabetes was also a contributing factor in all three cases [26]. In a second, larger study including 299 patients with advanced or recurrent disease not amendable to local therapy, the dose of medroxyprogesterone acetate was assessed. Patients were randomized to receive oral MPA, either 200 mg/d or 1000 mg/d. The low-dose regimen had a higher response rate than the high-dose regimen (ORR 25% vs. 15%, respectively). Median PFS and OS for the low-dose arm was 3.2 months and 11.1 months, respectively. The adjusted relative odds of responding to the high-dose regimen compared with the low-dose regimen was 0.61, making MPA 200 mg daily the preferred regimen. Prognostic factors which had a significant impact on the probability of response included initial performance status, age, histologic grade, and progesterone receptor status. Estrogen receptor status was not found to be significant. The most frequent adverse event was thrombophlebitis (5%). Pulmonary emboli were observed in 1% of patients [27].

## 3. Progestational Agents Combined with Tamoxifen

In order to improve the modest responses to single-agent hormonal therapy and overcome hormonal resistance thought to be secondary to the down-regulation of progesterone receptors, combination hormonal therapy has been utilized. Selective estrogen receptor modulators (SERMs) lead to increased progesterone receptor expression, making them attractive for combination therapy [18]. Pandya et al. attempted to compare single-agent megestrol acetate with the combination of megestrol acetate with the selective estrogen receptor modulator, tamoxifen (both given continuously). However, the megestrol acetate-alone arm was closed early due to poor accrual, and subsequently, they were unable to make a direct comparison between arms. The response rate on the megestrol plus tamoxifen arm with 42 eligible cases was found to be similar to single-agent progesterone at 19%, with a median survival of 8.6 months. There were more toxic complications observed on the combination arm, including a life-threatening case of pulmonary embolism [28].

The GOG conducted two studies utilizing alternating treatment strategies to take full advantage of progesterone receptor upregulation by tamoxifen. The first study evaluated MPA (200 mg/d) and tamoxifen (40 m/d) given every other week in 61 patients with recurrent or metastatic disease not amendable to local therapy. The overall response rate was 33% (10% complete response rate), with median PFS of 3 months and median OS of approximately 13 months [29]. In this study, ER was found to be statistically significantly in relation to clinical response [30]. In a similar protocol, MA (160 mg/d) was alternated with tamoxifen (40 mg/d) in a 3-week alternating sequence in 61 patients with measurable recurrent or advanced disease without prior cytotoxic or hormonal treatment. The protocol achieved a similar overall response rate of 27% but a higher complete response rate of 21%. When looking at the FIGO grade 1 tumor subgroup, the response rate increased to 38%. The median response duration was 28 months [31]. In both studies, weight gain was the most common adverse event, and thromboembolic disorders were the most common serious adverse events. The rate of grade 3 or 4 thromboembolic events was 2–8% [29,31]. Although the response rates are higher when alternating with tamoxifen, the benefit to survival is less clear [32].

## 4. Estrogen Receptor Agents Alone

Single-agent tamoxifen has been studied as one of the earliest non-progesterone, hormonally active agents in recurrent endometrial cancer, primarily in patients who had progressed on prior progesterone therapy. In an Australian study, single-agent tamoxifen was evaluated in 49 patients and found to have an ORR of 20%. The median overall survival of responders was 34 months, and the toxicity was minimal [33]. The role of single-agent tamoxifen was further assessed by the GOG in a larger study with 68 patients with advanced or recurrent disease with no prior systemic therapy, which found an ORR of only 10%. Similar to the progestins, responses occurred more frequently among patients with well-differentiated tumors. The median duration of PFS was 1.9 months, and the median duration of OS was 8.8 months. Again, the toxicity was minimal [34]. Given that tamoxifen is a weak estrogen agonist in the endometrium, tamoxifen is rarely prescribed as a single agent. A more pure estrogen receptor antagonist, fulvestrant, which is a selective estrogen receptor degrader (SERD), was also evaluated in endometrial cancer [35]. In a pair of studies, one by the GOG and one through the Arbeitsgemeinschaft Gynäkologische Onkologie (AGO), single-agent fulvestrant was found to have disappointing ORRs of 9.4% and 11.4%, respectively [36,37].

## 5. Aromatase Inhibitors

Aromatase is an important hormonal enzyme that catalyzes the peripheral conversion of androgens to estrogens [38]. Aromatase inhibitors (AIs) serve to block this peripheral conversion and therefore downregulate the production of estrogen and subsequent activation of estrogen receptors [38]. Single-agent aromatase inhibitors have been found to have modest activity for the treatment of recurrent endometrial cancer in multiple clinical trials as a second-line hormonal therapy option. The GOG evaluated single-agent anastrozole in 23 patients with no more than one prior hormonal therapy regimen and no prior chemotherapy, and it was found that the overall response rate was only 9% with no complete responses noted [39]. Single-agent letrozole was then evaluated in a Canadian trial in a similar cohort of 28 patients, with a similar overall response of 9.4%. Evaluation of biomarkers including ER did not show any correlation with response or disease progression [40]. Letrozole as a single agent was evaluated in the control arm of the PALEO study with a disease control rate of 38% at 24 weeks (ORR not yet reported) [41]. In each study, letrozole was well tolerated with very rare grade 3 and 4 events. Exemestane, an alternative AI, was evaluated by the Nordic Society of Gynecological Oncology (NSGO) as a single agent in 51 patients with advanced or recurrent disease stratified by ER status. They found a similar ORR of 10%, with no responses in the ER-negative group (ER-negative arm was closed early). In the ER-positive group, PFS was 3.8 months and OS 13.3 months. Again, treatment was well tolerated [42]. A summary of hormonal therapy options with efficacy results can be found in Table 1. 

## 6. Aromatase Inhibitors Combined with Targeted Therapies

Given the modest activity of aromatase inhibitors alone, they have been evaluated in combination with other therapeutic agents to improve efficacy. Specifically, AIs have been found to be complementary with targeted therapies that take advantage of downstream molecular pathways. In order to leverage the well-established crosstalk between the PI3K/AKT/mTOR pathway and hormonal signaling [22], the combination of letrozole and everolimus, an oral rapamycin analog that acts by selectively inhibiting mTOR, was evaluated in 38 patients with progressive or recurrent disease with up to two prior cytotoxic regimens, and it was found to have an ORR of 32% and a CBR of 40%. Patients both with and without ER expression were found to have a response to treatment [45]. This combination was also studied with the addition of a third drug, metformin, without apparent benefit [46]. The combination of letrozole and everolimus was further evaluated in comparison with the combination of MPA and tamoxifen (in a weekly alternating schedule) in a randomized controlled trial with 37 patients in each arm, with no more than one prior line of systemic chemotherapy. Letrozole and everolimus had an ORR of 22% compared with 25% in the MPA and tamoxifen group, with the highest response rates seen in chemo-naive patients (53% and 43%, respectively). Interestingly, chemo-naive patients also demonstrated the most durable response to letrozole and everolimus (28-month PFS compared to 5 months with MPA and tamoxifen) [43]. With the everolimus and letrozole combination, the most common toxicities were anemia, oral mucositis, hyperglycemia, fatigue, nausea, hypertriglyceridemia, and hypercholesterolemia. No grade 3 thromboembolic events were reported with this combination [43,45]. Of note, the combination of an mTOR inhibitor (temserolimus) with alternating MA and tamoxifen has also previously been evaluated in a clinical trial, however the combination arm was closed early due to an excess of venous thromboembolic events [47]. Most recently, an analogous phase I/II randomized study in France evaluated a different mTOR inhibitor, vistusertib, in combination with anastrozole in 49 heavily pretreated patients (one prior chemotherapy regimen and/or two lines of hormonal therapy except AI), who were all hormone receptor-positive. An ORR of 24.5% was reported in the combination arm compared with an ORR of 17.4% in the anastrozole alone arm. Median PFS was 5.2 months in the combination arm and 1.9 months in the anastrozole arm. Fatigue, lymphopenia, hyperglycemia, and diarrhea were the most common (grade ≥ 2) adverse events associated with vistusertib [44].

In addition to mTOR inhibitors, aromatase inhibitors are now being evaluated in combination with another class of targeted therapies, CDK4/6 inhibitors. CDK4/6 inhibitors have been studied extensively in breast cancer and have already gained regulatory approval. The first study published in endometrial cancer with the combination of letrozole and ribociclib, a third generation CDK4/6 inhibitor, showed that out of 20 ER+ endometrial cancer patients, 55% of patients were still on treatment at 12 weeks and 35% at 24 weeks. A total of 60% of patients had at least one grade 3 adverse event, the most common being bone marrow suppression and fatigue [48]. Recently published results from a second phase II trial evaluating letrozole with abemaciclib, an alternative CDK4/6 inhibitor, in 30 patients with recurrent or metastatic disease of all histologies with ER expression and no limit to prior therapies (cytotoxic or hormonal) found an ORR of 30% in a population of patients, of which 50% had prior hormonal therapy. The median duration of response was 7.4 months. The most common serious adverse events were due to bone marrow suppression [24]. A third randomized phase II trial by the NSGO has been presented but not yet published. They evaluated a third CDK4/6 inhibitor, palbociclib, in combination with letrozole (36 patients) and compared to letrozole alone (37 patients) in patients with ER+ tumors, with no limit to prior chemotherapy and one or fewer lines of hormonal therapy. They found that the combination therapy arm significantly improved PFS (8.3 months vs. 3.0 months). The disease control rate at 24 weeks was 64% for the combination arm (compared with 38% for letrozole alone). Similar to the other studies, the most common severe adverse events were due to bone marrow suppression [41]. A summary of combination hormonal with targeted therapy options with efficacy results can be found in Table 2.

## 7. Biomarkers of Response to Hormonal Therapy

Grade has long been the main predictive factor for response to hormonal therapy, with grade 1 tumors being most responsive and grade 3 being least responsive [26]. More recently, different molecular profiles have been identified and there is increasing emphasis on molecular profile over grade (and possibly even stage). Retrospective data has revealed ER status as a possible relevant biomarker in the NSMP molecular subgroup defining a group of low-risk patients [17]. In the phase II single arm study evaluating everolimus and letrozole, none of the patients with serous histology responded. In contrast, endometrioid histology had a 58% clinical benefit rate. When evaluating response based on ER status, both patients with and without ER expression had responses. Furthermore, molecular analysis revealed patients with CTNNB1 mutations had a clinical benefit rate of 80% (4/5), which increased to 100% (4/4) in patients with both endometrioid history and CTNNB1. However, many patients (54%) without CTNNB1 mutations also responded, suggesting further biomarkers need to be identified. In addition to CTNNB1, the investigators also evaluated KRAS and PIK3CA mutations, none of which were significantly correlated with response [45]. Responses were similarly evaluated based on molecular profile in Konstantinopoulos’ study evaluating letrozole and abemaciclib. The investigators observed responses in the trial regardless of grade, prior hormonal therapy, mismatch repair, and progesterone receptor status (all patients had ER expression by trial design). When evaluating responses based on molecular subtypes, they found responses in all subtypes except in the copy-number high (p53 abnormal) group and recommended future studies to further evaluate possible predictors of response, such as mutations of CTNNB1, KRAS, and CDKN2A [24].

At this time, in addition to molecular classification, guidelines recommend estrogen receptor testing in the setting of advanced-stage or recurrent disease [49,50]. Given the promising results that have been shown with the combination of hormonal therapy and targeted agents, there is much excitement about future possibilities, but no definite guidelines have been established for specific molecular testing. We anticipate this will change in the coming years based on ongoing clinical research. Further combination therapies are being investigated in ongoing clinical trials with AIs, including using CD4/6 inhibitors with or without the addition of mTOR inhibitors [51,52] as well as using CDK4/6 inhibitors in combination with an estrogen receptor antagonist (fulvestrant) [53]. Additional combinations are currently being studied, utilizing alternative pathway targets including PI3K inhibitors and AKT inhibitors [54,55,56]. A list of ongoing combination hormonal clinical trials is shown in Table 3.

## 8. Conclusions

Hormonal therapy has long been an integral treatment modality and alternative to cytotoxic chemotherapy for endometrial cancer. Despite the long history, advances continue to improve the hormonal treatment options available to patients by increasing response rates, extending the duration of response, and overcoming resistance mechanisms. Though traditional hormonal therapy (progesterones and tamoxifen) will continue to be an option for many patients (especially those that cannot tolerate or have contraindications to targeted therapies), the introduction of molecular classification has begun to transform the management of endometrial cancer. With this transformation, hormonal therapy will not only continue to be an important therapeutic option, but will be a vital component of future targeted therapy combinations as we have already started to see with mTOR inhibitors and CDK4/6 inhibitors. There are ongoing clinical trials to maximize the activity of combination therapies. The future of systemic treatment for recurrent endometrial cancer, like many cancers, is based on individualized targeted treatments based on molecular markers of each patient and each tumor. Based on what has been discovered regarding the molecular and genetic makeup of endometrial cancers, targeted therapies beyond mTOR and CDK4/6 have the possibility to increase activity when combined with hormonal therapy (e.g., MEK inhibitors and tyrosine kinase inhibitors [57]). High disease control rate, long duration of response, and easy tolerability make hormonal therapy an excellent option for patients with low-grade hormone receptor-positive disease who would like to avoid or cannot tolerate cytotoxic chemotherapy or other targeted therapy. In a time of a growing number of therapeutic options for our patients with recurrent endometrial cancers, one should never forget about the option of hormonal therapy.

## Figures and Tables

**Table 1 cancers-15-01799-t001:** Hormonal therapy options with efficacy results.

Agent (Dose/Schedule)	ORR	CBR	PFS (m)	OS (m)	DOR (m)	Study
Single-agent Progesterone						
Megestrol acetate (MA) (800 mg/d)	24%	46%	2.5	7.6	8.9	Lentz et al., 1996 [26]
Medroxyprogesterone Acetate (MPA) (200 mg/d)	25%		3.2	11.1		Thigpen et al., 1999 [27]
Progesterone and Aromatase Inhibitor						
MA (160 mg) and tamoxifen (20 mg/d), continuous	19%			8.6		Pandya et al., 2001 [28]
MA (160 mg/d) and tamoxifen (40 mg/d), alternating every 3 weeks	27%		2.7	14		Fiorica et al., 2004 [31]
MPA (200 mg/d) and tamoxifen (40 mg/d), alternating weekly	33%		3	13		Whitney et al., 2004 [29]
MPA (200 mg/d) and tamoxifen (40 mg/d), alternating weekly	25%	69%	4	17	31	Slomovitz et al., 2022 [43]
Single-agent SERM/SERD						
Tamoxifen (40 mg/d)	10%		1.9	8.8	1.9	Thigpen et al., 2001 [34]
Fulvestrant (250 mg IM every 4 w)	9%	34%	2	18.9		Covens et al., 2011 [36]
Fulvestrant (250 mg IM every 4 w)	11%	34%	2.3	13.2		Emons et al., 2013 [37]
Single-agent Aromatase Inhibitor						
Anastrozole (1 mg/d)	9%	17%	1	6		Rose et al., 2000 [39]
Anastrozole (1 mg/d)	17%		1.9			Heudel et al., 2022 [44]
Examestane (25 mg/d)	10%	35%	3.1	10.9		Lindemann et al., 2014 [42]
Letrozole (2.5 mg/d)	9%	44%	3.9	8.8	6.7	Ma et al., 2004 [40]

ORR: objective response rate, CBR: clinical benefit rate, PFS: median progression free survival, OS: median overall survival, DOR: median duration of response. Blank results indicated data not available.

**Table 2 cancers-15-01799-t002:** Combination hormonal with targeted therapy with efficacy results.

Agent (Dose/Schedule)	ORR	CBR	PFS (m)	OS (m)	DOR (m)	Study
Aromatase Inhibitor and mTOR Inhibitor						
Letrozole (2.5 mg/d) and Everolimus (10 mg/d)	32%	40%	3	14		Slomovitz et al., 2015 [45]
Letrozole (2.5 mg/d) and Everolimus (10 mg/d)	22%	78%	6	31	30	Slomovitz et al., 2022 [43]
Anastrazole (1 mg/d) and Vestusertib (250 mg, 2 d per week)	25%		5.2			Heudel et al., 2022 [44]
Aromatase Inhibitor and CDK4/6 Inhibitor						
Letrozole (2.5 mg/d) and Ribociclib (400 mg/d)	10%		5.4	15.7		Colon-Otero et al., 2020 [48]
Letrozole (2.5 mg/d) and Abemaciclib (300 mg/d)	30%	73%	9.1	21.6	7.4	Konstantinopoulos et al., 2022 [24]
Letrozole (2.5 mg/d) and Palbociclib (125 mg/d)		64%	8.3			Mirza et al., 2020 [41]

ORR: objective response rate, CBR: clinical benefit rate, PFS: median progression free survival, OS: median overall survival, DOR: median duration of response. Blank results indicated data not available.

**Table 3 cancers-15-01799-t003:** Combination hormonal therapy with targeted therapy, ongoing clinical trials.

Class.	Study Title	Design	Sponsor	Estimated Completion Date	NCT Number
Aromatase Inhibitor and mTOR Inhibitor, +/− CDK4/6 Inhibitor	A Phase II, Two-Arm Study of Everolimus and Letrozole, +/− Ribociclib (Lee011) in Patients With Advanced or Recurrent Endometrial Carcinoma	Phase II, two arm	M.D. Anderson Cancer Center	31 August 2023	NCT03008408
Aromatase Inhibitor and CDK4/6	Abemaciclib and Letrozole to Treat Endometrial Cancer	Phase II, single arm	GOG	1 September 2024	NCT04393285
SERD and CDK4/6 Inhibitor	Evaluating Cancer Response to Treatment With Abemaciclib and Fulvestrant in Women With Recurrent Endometrial Cancer	Phase II, single arm	Memorial Sloan Kettering Cancer Center	August 2023	NCT03643510
SERD and PI3K Inhibitor	A Study of Alpelisib and Fulvestrant to Treat Endometrial Cancer	Phase II, single arm	GOG	1 April 2026	NCT05154487
SERD and PI3K Inhibitor	Phase 2 Study of PI3K Inhibitor Copanlisib in Combination With Fulvestrant in Selected ER+ and/or PR+ Cancers With PI3K (PIK3CA, PIK3R1) and/or PTEN Alterations	Phase II, dose expansion	M.D. Anderson Cancer Center	29 October 2026	NCT05082025
Progesterone and AKT Inhibitor	Testing the Addition of the AKT Inhibitor, Ipatasertib, to Treatment With the Hormonal Agent Megestrol Acetate for Recurrent or Metastatic Endometrial Cancers	Phase IB/Phase II	National Cancer Institute/NRG Oncology	31 January 2027	NCT05538897

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
