# Peer review of "Do Not Forget about Hormonal Therapy for Recurrent Endometrial Cancer: A Review of Options, Updates, and New Combinations"

_cancers, 2023, doi:10.3390/cancers15061799_

Round 1
Reviewer 1 Report
Comment on “Don't forget about hormonal therapy for recurrent endometrial cancer: a review of options, updates, and new combinations”:
Authors have reviewed the role of hormonal therapy in managing recurrent endometrial cancers. It is a topic of interest and is an unmet need with ongoing advancements in molecular subtyping of endometrial cancers. Review reads well.
Addressing following points may help further.
1. It would be better, if indication for hormonal therapy are explained bit more clearly. Although it is mentioned as recurrent endometrial cancer broadly, the review does not highlight the prior treatment modalities in the studies mentioned. This could help clinicians to select suitable cases over RT/chemotherapy.
2. A section on need for testing of metastatic disease for hormonal receptors and evidence regarding this can be added
3. Au : may add a paragraph on future directions in recurrent endometrial cancers for hormonal therapy? Expert opinion can be added regarding foreseeable changes in recurrent endometrial cancer such as the role of molecular profiling in influencing patient selection for hormonal therapy.
4. In lines 52-56 of page 2: Authors state that all the molecular groups of endometrial cancer except for CNH express high levels of hormone receptors and quote study by Levine et al. But hormone receptor are highly expressed in CNL group, while CNH group shows very low levels of expression. It cannot be generalized that the other two groups also show high levels of ER/PR expression.
5. In lines 57-59, the authors state that association of ER expression with survival outcomes is controversial. But the studies referenced do not provide evidence to this statement (Rodriguez et al). Can the authors please elaborate more about this statement?
6. Ref 15 doesn’t seem to reference downstream signaling of PI3K/Akt/mTOR pathway in endometrial cancers
7. Number of cases in each study included in review should be mentioned in detail if available.
8. Is it possible to get the receptor status from available studies affecting the choice of hormonal therapy. What shall be the advice to clinicians regarding routine receptor testing prior to consideration for hormonal therapy as receptor status may vary in primary and recurrent disease.
9. Thromboembolic complications in studies evaluating progestational agents alone, progestational agents with tamoxifen, estrogen receptor agents alone should be discussed more along with numbers affected as they were most common serious adverse events.
10. With promising outcomes seen with targeted agents combined with AI, do the authors consider a role of single agent progesterone, estrogen receptor agents? Conclusion of the review can address this in more detail.
Author Response
"Please see the attachment."

Reviewer 2 Report
The article about hormonal treatment for advanced or recurrent endometrial cancer was extremely interesting and well-written. This article is informative for an audience of clinicians and deserves to be considered for publication. However, it required substantial revision before achieving further consideration. According to the SANRA scale, an adequate methods section must be included: Baethge C, Goldbeck-Wood S, Mertens S. SANRA-a scale for the quality assessment of narrative review articles. Res Integr Peer Rev. 2019;4:5.
Author Response
"Please see the attachment."

Round 2
Reviewer 2 Report
The authors adequately addressed the issues raised.